# RE-SORT: Removing Spurious Correlation in Multilevel Interaction for CTR Prediction

**Song-Li Wu**[‡*1]    **Liang Du**[‡2]    **Jia-Qi Yang**[1]    **Yu-Ai Wang**[3]    **De-Chuan Zhan**[1]    **Shuang Zhao**[2]    **Zi-Xun Sun**[†2]

[1] National Key Laboratory for Novel Software Technology, Nanjing University, China
[2] Interactive Entertainment Group, Tencent Inc., China
[3] Fudan University, China

## Abstract

Click-through rate (CTR) prediction is a critical task in recommendation systems, serving as the ultimate filtering step to sort items for a user. Most recent cutting-edge methods primarily focus on investigating complex implicit and explicit feature interactions; however, these methods neglect the spurious correlation issue caused by confounding factors, thereby diminishing the model's generalization ability. We propose a CTR prediction framework that **RE**moves **S**purious c**OR**relations in mul**T**ilevel feature interactions, termed **RE-SORT**, which has two key components. I. A multilevel stacked recurrent (MSR) structure enables the model to efficiently capture diverse nonlinear interactions from feature spaces at different levels. II. A spurious correlation elimination (SCE) module further leverages Laplacian kernel mapping and sample reweighting methods to eliminate the spurious correlations concealed within the multilevel features, allowing the model to focus on the true causal features. Extensive experiments conducted on four challenging CTR datasets and our production dataset demonstrate that the proposed method achieves state-of-the-art performance in both accuracy and speed. The codes will be released at https://github.com/RE-SORT.

## 1 INTRODUCTION

Click-through rate (CTR) prediction holds significant importance in recommendation systems [Fan et al., 2023, Ye et al., 2022, Rendle et al., 2009] and online advertising scenarios. The objective of this task is to gauge the likelihood of a user

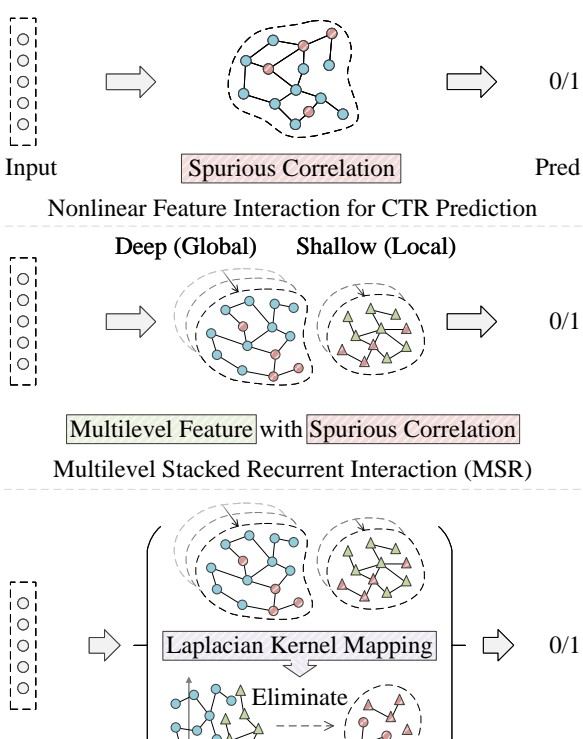

Figure 1: RE-SORT exploits multilevel features with spurious correlation elimination for CTR prediction.

clicking on a recommended item or an advertisement displayed on a web page. Developing more effective methods for modeling user and item features has emerged as a crucial research challenge in the CTR prediction field.

To learn comprehensive feature interactions, various models have been developed. On the one hand, Xiao et al. [2017], Rendle [2010], Pan et al. [2018], and Sun et al. [2021] learn low-order feature interactions to achieve low temporal and spatial complexity. However, relying solely on first-order and second-order feature interactions results in limited per-

---

*This work was done during his internship at Tencent.
†Corresponding authors: zixunsun@tencent.com.
‡Indicates equal contribution.

formance [Juan et al., 2016]. On the other hand, models such as AutoInt [Song et al., 2019] and SAM [Cheng and Xue, 2021] employ self-attention mechanisms to capture high-order feature interactions, thereby enhancing their representation capabilities. To further incorporate richer high-order features, two-stream CTR models that leverage dual parallel networks to capture feature interactions from distinct perspectives (e.g., explicit vs. implicit perspectives) have been developed. For instance, the DCN [Wang et al., 2017] automates the feature cross-coding process through linear and nonlinear feature learning streams. Building upon the DCN, the GDCN [Wang et al., 2023] introduces a soft gate to the linear feature learning stream to filter important features. In contrast with the DCN and GDCN, FinalMLP [Mao et al., 2023] uses two nonlinear feature learning streams to implicitly encode the features at distinct levels by setting multilayer perceptrons (MLPs) with different parameters.

While the existing two-stream methods have achieved superior performance by diversifying the features through the introduction of various feature interaction streams, there is still substantial room for improvement in implicitly constructing hierarchical features within each stream (please refer to Section 4.3 for the experimental validation). Previous works such as [Du et al., 2021] and [Pascanu et al., 2013] have demonstrated that stacked recurrent structures can learn more intricate representations compared to the feed-forward structures such as MLP structures, allowing each layer to potentially represent different levels of abstraction. By restoring the previous state in the recurrent structure [Quadrana et al., 2017], the use of a greater number of stacks leads to better integrated global high-level representations (e.g., the coarse-grained common patterns of the user preferences for recommended items), while the use of fewer stacks results in increased feature combinations with local details (e.g., the fine-grained associations between users and items). To more effectively differentiate among the hierarchical distinctions of the features in each stream, we propose a two-stream multilevel stacked recurrent (MSR) structure that leverages the capacity of the network to capture both global and local features. Based on the proposed stacked recurrent structure, an acceleration module is further introduced to substantially boost the inference efficiency of the proposed approach.

Furthermore, although the two-stream structure can extract richer features, as illustrated in Figure 1 (rows 1 and 2), it is still hindered by the presence of "spurious correlations", which refer to statistical relationships between two or more variables that appear to be causal but are not causal. These spurious correlations inherently arise from the subtle connections between noisy features and causal features [Li et al., 2022], which reduces the model's generalization ability [Lu et al., 2021]. For example, in movie recommendation tasks, the prominence of certain trending films may result in higher click counts due to their prioritized placements. However,

the actual preferences of users might not align with the genres or contents of these trending movies. This creates spurious correlations between the popularity of certain film types and the genres that users truly appreciate, and movie placement is the confounding factor. Recent studies [Mao et al., 2023, Wang et al., 2021a, Guo et al., 2017] have confirmed through meticulous parameter analyses that model performance decreases when the interaction order exceeds a certain depth, typically three orders [Wang et al., 2023], and one of the crucial reasons for this is the exacerbation of the spurious correlation issue. Previous works [Wang et al., 2021a, 2023] have used gating mechanisms to assign varying levels of importance to different features. However, in the absence of a supporting causal theory [Guan et al., 2023], feature selection methods fail to identify the true causal features, instead favoring features that arise from spurious correlations. LightDIL [Zhang et al., 2023] divided historical data into multiple periods chronologically, forming a set of environments, and learned stable feature interactions within these environments, yet its effectiveness diminishes when users have constrained click histories.

As shown in Figure 1 (row 3), to address the spurious correlation issue, we employ the Laplacian kernel function to project low-dimensional feature interactions into a high-dimensional space. Thus, nonlinear transformations can be achieved in the low-dimensional space through a linear transformation in the high-dimensional space. Then, we use a sample reweighting strategy that learns different weights for various instances during training to eliminate spurious correlations. The detailed theory is explained in Section 3.5. Leveraging the combined power of the MSR and SCE, we have exploited the network's ability to eliminate the spurious correlations concealed within the hierarchical feature spaces, which enhances the CTR prediction process. The contributions are summarized as follows:

- We propose a CTR prediction framework that removes spurious correlations in multilevel feature interactions; this approach leverages the hierarchical causal relationships between items and users to fundamentally enhance the model's generalization ability.

- We propose a multilevel stacked recurrent (MSR) structure, which efficiently builds diverse feature spaces to obtain a wide range of multilevel high-order feature representations.

- We introduce a spurious correlation elimination (SCE) module, which utilizes Laplacian kernel mapping and sample reweighting methods to eliminate the spurious correlations hidden in multilevel feature spaces.

- The results of extensive experiments conducted on four challenging CTR datasets and our production dataset demonstrate that the proposed RE-SORT achieves state-of-the-art (SOTA) performance in terms of both accuracy and speed.

## 2 RELATED WORKS

### 2.1 RECURRENT STRUCTURE

Recurrent structures have found widespread application in various research fields, including computer vision (CV) and natural language processing (NLP). For instance, in stereo matching, a recurrent hourglass network is proposed by Du et al. [2021] to effectively capture multilevel information. This strategy enhances the adaptability of the network to complex scenarios by leveraging comprehensive global and local features. In the realm of recommendation systems, DRIN [Jun et al., 2022] employs a recurrent neural network (RNN) with $1 \times 1$ convolution kernels to recurrently model the second-order feature interactions between a raw feature and the current feature. However, DRIN utilizes only a single vanilla RNN branch, overlooking the potential benefits of a two-stream structure with varying stacking depths for simultaneously capturing both global and local feature representations. In contrast, our two-stream MSR structure is equipped with a self-attention mechanism, attenuation units, and an acceleration method. This design fundamentally enhances the capacity of the network to efficiently capture multilevel high-order features.

### 2.2 SPURIOUS CORRELATION IN RECOMMENDATION TASKS

In the field of causal inference, causality refers to the cause-effect relationship between variables, where changes in one variable trigger changes in another variable. The objective is to identify these relationships. Conversely, a spurious correlation refers to an observed association that does not necessarily indicate a genuine causal connection, which often occurs due to the presence of confounding variables or selection bias [Spirtes et al., 2013]. Dou et al. [2022] utilized feature decorrelation to promote feature independence, thereby mitigating spurious textual correlations for natural language understanding. Zhang et al. [2021] employed sample reweighting for variable decorrelation tasks to identify stable features for image classification. In recommendation tasks, Li et al. [2022] incorporated causal feature selection into factorization machines (FMs) to address the issue of confounding factors and selection bias, thereby improving the robustness of the output recommendations. They learned the weights associated with each first-order and second-order feature to explicitly select causal features. However, not all causal features are limited to these orders, and the employed optimization method hinders efficiency, particularly when handling many features. In contrast, our proposed method not only learns the weight of every instance to select causal features and feature interactions with arbitrary orders but also boasts an efficient structure with a rapid inference speed, providing a more comprehensive approach.

## 3 METHOD

In this section, we introduce the proposed MSR and SCE, the CTR predictor, the loss function, and the theory behind the sample reweighting method in the SCE module.

### 3.1 MULTILEVEL STACKED RECURRENT (MSR) INTERACTION

The adoption of a two-stream model structure for CTR prediction stems from insights gleaned from prior research and practical applications [Wang et al., 2023, Mao et al., 2023, Wang et al., 2021b,a]; this structure helps enhance the expressiveness of the constructed model. Following previous works, the proposed MSR method contains two streams: a deeply stacked recurrent (D-SR) stream and a shallow stacked recurrent (S-SR) stream. As shown in Figure 2, each stream includes multiple stacked recurrent blocks with a self-attention structure and varying attenuation coefficients, thus enhancing the depth-based pattern and dependency learning process. The input feature of both D-SR and S-SR is denoted as $x_0$, and the outputs of D-SR and S-SR are denoted as $F_d$ and $F_s$, respectively. The calculation process is defined as: $F_d = \text{D-SR}(x_0)$, $F_s = \text{S-SR}(x_0)$. We take the i-th block of the D-SR as an example. Given an input $x_{i-1}$, we first project it into a one-dimensional function $w_v$: $V_i = x_{i-1} \cdot w_v$ and then map $V_i$ to $x_i$ through the state $S_i$. We recurrently calculate the output as follows:

$$S_i = r_i S_{i-1} + K_i^T V_i, \tag{1}$$

$$x_i = \text{SR}_i(x_{i-1}) = Q_i S_i, \tag{2}$$

where $\text{SR}_i$ denotes the i-th block, $Q_i$, $K_i$, and $V_i$ are projections, $r_i$ is the attenuation coefficient, and $i \in 1, 2, 3, ..., M$. We further add a swish gate [Ramachandran et al., 2017] in each stream to increase the non-linearity of MSR layers.

**Enhancing computational efficiency in MSR.** In addition, we find that many weights in the softmax function are applied to abnormal or null values, which results in unnecessary computational costs. Therefore, we propose changing the softmax function to learnable matrices by using a relative position encoding [Sun et al., 2022] in the stacked recurrent structure to reduce the incurred temporal and spatial costs.

Formally, we construct the layer as shown in Algorithm 1, where $P_Z$ and $P_U$ are learnable matrices used to replace the softmax function, and "GroupNorm" [Shoeybi et al., 2019] normalizes the output of each block. In summary, the different depths of the D-SR and S-SR streams lead to significant differences among the learning feature interactions of any order, enabling the model to grasp the global and local dependencies at various abstraction levels.

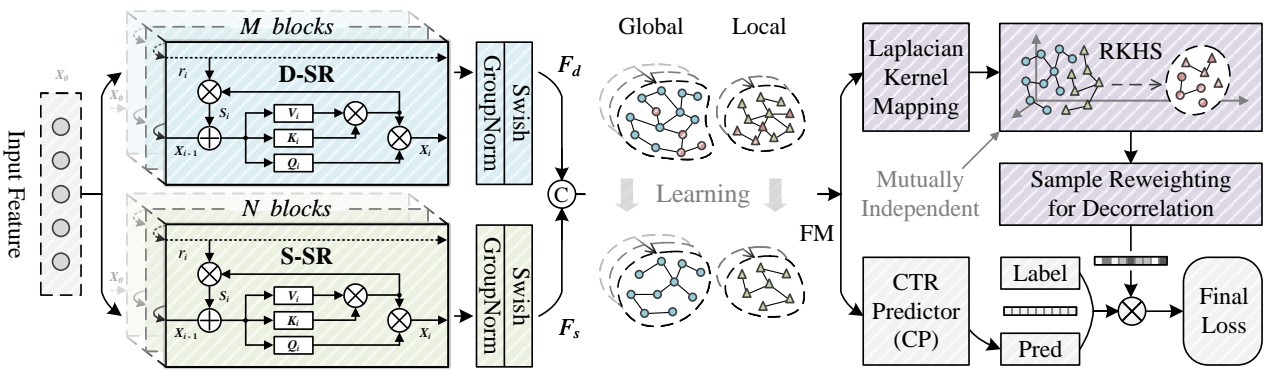

Figure 2: Overview of the RE-SORT framework, which contains two key components: I. a multilevel stacked recurrent (MSR) structure and II. a spurious correlation elimination (SCE) module. The deep and shallow stacked recurrent interactions (D-SR and S-SR) in the MSR structure and the SCE module are colored blue, green, and purple, respectively. The D-SR and S-SR streams contain M and N (M > N) stacked blocks, and "$Q_i$", "$K_i$", and "$V_i$" denote the query, key, and value, respectively, for $i$-th block. The attenuation coefficient is denoted by $r_i$ for $i$-th block (dotted lines). The outputs $F_d$ and $F_s$ are concatenated ("C") as the input of the SCE and the CTR predictor, which is visualized to show the spurious correlation elimination process guided by the SCE module. "RKHS" denotes the reproducing kernel Hilbert space. The SCE procedure is explained in Algorithm 2.

---

**Algorithm 1** The procedure of D-SR

---

**Input:** Input data $x_0$, Attenuation coefficient $r_i$, Learnable matrices $P_Z$, $P_U$, Blocks number: $M$
**Output:** $F_d$
1: $x_1 \leftarrow \text{SR}_1(x_0, r_1)$
2: **for** i $\leftarrow$ 2 **to M do**
3:     $x_i \leftarrow \text{SR}_i(x_{i-1}, r_i)$
4: **end for**
5: $T \leftarrow \text{GroupNorm}([x_1, x_2, ..., x_M])$
6: $F_d \leftarrow (\text{swish}(x_0 \cdot P_Z) \odot T) \cdot P_U$
7: **return** $F_d$

---

### 3.2 SPURIOUS CORRELATION ELIMINATION

Since we've established distinct feature spaces with the MSR, our next objective is to eliminate the spurious correlations concealed within them. We concatenate the last two outputs $F_d$, $F_s$ to form a local feature map FM.

**Statistical independence assessment.** To eliminate spurious correlations, we use sample reweighting to make the spurious correlations independent of the user click behavior prediction task $y_i$. The underlying theory is elaborated upon in Section 3.5 and the appendix. Specifically, inspired by the kernel function in a support vector machine (SVM), we map the features into a high-dimensional space with a Laplacian kernel. In the high-dimensional space, we eliminate the spurious correlations between features with nonlinear operations corresponding to the original feature space. We use $X$ and $Y$ to denote the features, and the SCE module measures the dependence between them. The theory regarding why

SCE measures dependence is also given in the appendix.

$$\text{SCE}(X, Y) = ||K_X - K_Y||_{\text{FN}}, \quad (3)$$

where $K_X = k_X(X, X)$ and $K_Y = k_Y(Y, Y)$ are Laplacian kernel matrices, $|| \cdot ||_{\text{FN}}$ is the Frobenius norm (FN) which corresponds to the Hilbert-Schmidt norm in Euclidean space [Strobl et al., 2019], and $k_X$ and $k_Y$ are Laplacian kernels that can capture local patterns and are robust to outliers. However, applying the SCE approach with large-scale kernel matrices can be computationally expensive. Therefore, we use random Fourier features (RFFs) to approximate the Laplacian kernel. Afterwards, the reconstructed features can be obtained in the new representation space, which reduces the temporal complexity of the network from $O(n^2)$ to $O(n)$.

$$\mathcal{H}_{\text{RFF}} = \{h : x \to \sqrt{2}\cos{(wx + \phi)}| \\ w \sim N(0, 1), \phi \sim U(0, 2\pi)\}, \quad (4)$$

where $\mathcal{H}_{\text{RFF}}$ denotes the function space of RFF, $\sqsupseteq$ is sampled from the standard normal distribution, and $\phi$ is sampled from the uniform distribution.

We define the partial cross-covariance matrix as the measure of covariance between the two sets of features:

$$\text{SCE}_{\text{RFF}}(X, Y) = ||C_{p(X), q(Y)}||_{\text{FN}}^2 \quad (5)$$
$$= \sum_{i=1}^{n_X} \sum_{j=1}^{n_Y} |\text{Cov}(p_i(X), q_j(Y))|^2, \quad (6)$$

where $X$ and $Y$ represent features, $p$ and $q$ are random Fourier feature mapping functions, $n_X$ and $n_Y$ are the number of functions from $\mathcal{H}_{\text{RFF}}$, $|| \cdot ||_{\text{FN}}$ is the Frobenius norm

**Algorithm 2** Procedure of SCE module

---

**Input:** Balancing epoch number, Batch size
**Output:** Trained model with weight $w_{\text{result}}$

1: Define: $n$: the number of random Fourier space
2: Initialize: sample weights: $w_{\text{result}} = [1, 1,..., 1]$
3: Reload global features and weights as Eq. (12), Eq. (13)

4: **for** batch $\leftarrow 1$ **to** Batch size **do**
5:    **for** epoch balancing $\leftarrow$ Balancing epoch number **to** 1 **do**
6:      $n_X = n_Y$ = epoch balancing
7:      Optimize $w_{\text{result}}$ with $n_X$ and $n_Y$ as Eq. (11)
8:    **end for**
9:    Back propagate with weighted prediction loss as Eq. (18)
10:   Save features and weights in GFI′ and GWI′ as Eq. (14), Eq. (15)
11: **end for**

---

(FN), and $C_{p(X),q(Y)} \in \mathbb{R}^{n_X \times n_Y}$ is the cross-covariance matrix of random Fourier features $p(X)$ and $q(Y)$ containing entries:

$$p(X) = (p_1(X), ..., p_n(X)), p_i(X) \in \mathcal{H}_{\text{RFF}}, \forall i, \quad (7)$$
$$q(Y) = (q_1(Y), ..., q_n(Y)), q_j(Y) \in \mathcal{H}_{\text{RFF}}, \forall j. \quad (8)$$

**Global sample weight optimization.** In this section, we present a method for optimizing the global features and global sample weights to enhance the performance of the model. Our approach aims to effectively capture features and assign appropriate weights to the samples. By iteratively updating the feature weights and incorporating global and local information, we can improve the feature representations and their integration into the model. By minimizing $\text{SCE}^w_{\text{RFF}}(X, Y)$, we encourage feature independence, resulting in a more causal and consistent covariate matrix. For any two features $X_{:,a}, X_{:,b} \in X$, the weighted spurious correlation elimination process is $\text{SCE}^w_{\text{RFF}}(X, Y)$.

$$\text{SCE}^w_{\text{RFF}}(X_{:,a}, X_{:,b}, w)$$
$$= \sum_{i=1}^{n_X} \sum_{j=1}^{n_Y} |\text{Cov}(p_i(w^T X_{:,a}), q_j(w^T X_{:,b}))|^2. \quad (9)$$

In each iteration, our objective is to minimize the sum of Eq. (9). Therefore, the resulting weight is

$$w_{\text{result}} = \arg\min_w \sum_{1 \le a \le b \le m} \text{SCE}^w_{\text{RFF}}(X_{:,a}, X_{:,b}, w) \quad (10)$$
$$= \arg\min_w \sum_{1 \le a \le b \le m} \sum_{i=1}^{n_X} \sum_{j=1}^{n_Y} |\text{Cov}(u_i, v_j)|^2, \quad (11)$$

where $u_i = p_i(w^T X_{:,a})$, $v_j = q_j(w^T X_{:,b})$. To avoid local optima, we incorporate global information. We employ a memory module consisting of GFI and GWI. GFI captures global feature information, while GWI stores global weight information. The features and weights are used to optimize the new sample weights.

$$\text{GFI}_i = \text{Concat}(\text{GFI}_{i-1}, \text{FM}_i), \quad (12)$$
$$\text{GWI}_i = \text{Concat}(\text{GWI}_{i-1}, w_i), \quad (13)$$

where $\text{GFI}_1 = \text{FM}_1$, $\text{GWI}_1 = w_1$, $\text{FM}_i$ denotes the local feature information, $w_i$ is the local weight information, and $i$ is means the number of iterations. We globally update the features and weights as follows:

$$\text{GFI}'_i = \frac{1}{2}(\text{GFI}'_{i-1} + \text{FM}_i), \quad (14)$$
$$\text{GWI}'_i = \frac{1}{2}(\text{GWI}'_{i-1} + w_i), \quad (15)$$

where $\text{GFI}'_1 = \frac{1}{2}\text{FM}_1$ and $\text{GWI}'_1 = \frac{1}{2}w_1$.

### 3.3 CTR PREDICTOR

We divide the $F_d$ and $F_s$ into $k$ chunks denoted as $F_d = [F_{d_1}, ..., F_{d_k}]$, $F_s = [F_{s_1}, ..., F_{s_k}]$, where $k$ is a hyperparameter. $F_{d_j}$ and $F_{s_j}$ denote the j-th chunk feature of the D-SR and S-SR outputs, respectively. We apply the CTR predictor (CP) to each paired chunk group consisting of $F_{d_j}$ and $F_{s_j}$. The chunk computations are then aggregated using sum pooling to derive the final predicted probability:

$$\text{CP}(F_{d_j}, F_{s_j}) = b + w_{d_j}^T F_{d_j} + w_{s_j}^T F_{s_j} + F_{d_j}^T W_j F_{s_j}, \quad (16)$$
$$\hat{y} = \sigma(\sum_{j=1}^k \text{CP}(F_{d_j}, F_{s_j})), \quad (17)$$

where $w_{d_j}$, $w_{s_j}$, and $W_j$ are learnable weights, and $\sigma$ is the sigmoid activation function. Modeling the second-order interactions between hierarchical features $F_{d_j}^T W_j F_{s_j}$ actually involves modeling arbitrary-order feature interactions.

### 3.4 LOSS FUNCTION

We introduce a reweighting scheme for each sample within the commonly utilized binary cross-entropy loss. This involves applying a distinct weight to the loss of each sample during training.

$$L = -\sum_{i=1}^N w_{\text{result}_i}(y_i \log(\hat{y}_i) + (1 - y_i)\log(1 - \hat{y}_i)), \quad (18)$$

where $N$ is the number of examples, $y_i$ is the true label of instance i, and $\hat{y}_i$ is the predicted probability of a click. During training, we use the weight $w_{\text{result}} = [w_{\text{result}_1}, w_{\text{result}_2}, ....w_{\text{result}_N}]$ of the $N$ LogLoss values calculated for every input sample with the sample reweighting mechanism to conduct gradient descent.

| Datasets | # Samples | # Fields | # Features |
|----------|-----------|----------|------------|
| Criteo | 45,840,617 | 39 | 2,086,936 |
| Avazu | 40,428,967 | 23 | 1,544,250 |
| Frappe | 288,609 | 10 | 5,382 |
| MovieLens | 2,006,859 | 3 | 90,445 |
| UGC | 1,015,236,921 | 92 | 10,640,310 |

Table 1: Statistics of the evaluation datasets.

## 3.5 THE THEORY BEHIND SAMPLE REWEIGHTING

In the following section, we denote the covariate features as FM=[FM$_{rob}$, FM$_{fal}$], where FM$_{rob}$ denotes the robust features, and FM$_{fal}$ = FM/FM$_{rob}$ denotes spurious correlation features. $g(\cdot)$ is a nonlinear function learned by the model with FM$_{rob}$ and FM$_{fal}$. $\beta_{FM_{rob}}$ and $\beta_{FM_{fal}}$ denote the linear weights of FM$_{rob}$ and FM$_{fal}$. He et al. [2023] demonstrate that $\beta_{FM_{fal}}$ asymptotically exceeds 0, implying that models without the SCE module inevitably suffer from the impact of spurious correlation features. Additionally, the correlation between FM$_{rob}$ and $g(FM_{rob})$ exhibits less variation than the correlation between FM$_{fal}$ and $g(FM_{rob})$ when exposed to parameter variance. Even when sample reweighting is applied randomly, the estimate of $\beta_{FM_{rob}}$ is more robust and less variable than the estimate of $\beta_{FM_{fal}}$. When predicting $y_i$, $y_i$ and FM$_{fal}$ must be statistically independent of the minimal and optimal predictor FM$_{rob}$: $y_i \perp FM_{fal}|FM_{rob}$. Xu et al. [2022] prove that FM$_{rob}$ is the minimal and optimal predictor if and only if FM$_{rob}$ is the minimal robust feature set. Therefore, we intend to capture the minimal robust feature set FM$_{rob}$ to obtain robust prediction results. We denote the robust feature set as the minimal robust feature set for simplicity. Let $\mathcal{W}$ be a set of sample weighting functions. Our objective is to acquire $\mathcal{W}_\perp$, a subset of $\mathcal{W}$ in which the features in FM are mutually independent. Zhou et al. [2022] prove that there exists a weight function $w \in \mathcal{W}$ that makes the spurious correlations independent of $y_i$ for linear models. Moreover, Xu et al. [2021b] prove that whether the data generation process is linear or nonlinear, conducting a weighted least squares (WLS) operation using the weighting function in $\mathcal{W}_\perp$ can lead to the selection of perfect features. Therefore, we transform the task from finding minimal robust features to obtaining an independent FM by sample reweighting.

## 4 EXPERIMENTS

### 4.1 DATASETS AND EVALUATION METRICS

We conduct extensive experiments on four official datasets and our production dataset. Table 1 provides an overview of the statistical information. We utilize two evaluation metrics, the area under the ROC curve (AUC) and LogLoss (cross-

entropy loss).

**Criteo.** Criteo is an online advertising and CTR prediction dataset that includes real-world displayed ads.

**Avazu.** Avazu is a widely used dataset in CTR prediction. This dataset includes real-world advertising data, focusing on mobile ads and user demographics.

**Frappe.** Frappe includes smartphone sensor data, which is popular in human activity recognition research.

**MovieLens.** MovieLens is used in recommender systems and leverages its movie ratings data to predict the likelihood of a user clicking on a movie.

**User Game Content (UGC).** The UGC dataset consists of clicking behavior data for user game content collected from our online service over one month. We collected clicking logs with anonymous user IDs, user behavior histories, game content features (e.g., categories), and context features (e.g., operation time).

### 4.1.1 Baseline

To fairly and accurately verify the improvements achieved by the proposed MSR structure and SCE module, we use a two-stream MLP network as the "Baseline" method. The MLP sizes of the two branches are set to [400, 400, 400] and [800] as these values yield the best performance. This approach aligns with established best practices for two-stream CTR models [Mao et al., 2023], providing a solid foundation for evaluating any modifications or novel approaches in a controlled manner. To further verify the superiority of the proposed recurrent structure, we utilize a self-attention structure to replace the MLP in the "Baseline" model and denote it as "Baseline + SA". The implementations of these models can be found in our open-source code.

### 4.1.2 Implementation

To conduct fair comparisons with the recently proposed cutting-edge methods, we use the released preprocessed datasets produced by Cheng et al. [2020] with the same splitting and preprocessing procedures. All the models and experiments are implemented based on the FuxiCTR toolkit [Zhu et al., 2021]. RE-SORT and the "Baseline" model follow the same experimental settings as those of FinalMLP [Mao et al., 2023], with the batch size for all datasets set to 4,096, the embedding size set to 10 for the five datasets, and the learning rate set to 0.001 unless otherwise specified. The number of chunks $k$ in the CP is set to 50 for Criteo and UGC, 10 for Avazu and MoiveLens, and 1 for Frappe. The remaining hyperparameters are kept constant for the five

---

http://labs.criteo.com
https://www.kaggle.com
https://github.com/openbenchmark/BARS

| Class | Model | Criteo | | Avazu | | MovieLens | | Frappe | |
|---|---|---|---|---|---|---|---|---|---|
| | | AUC ↑ | Logloss ↓ | AUC ↑ | Logloss ↓ | AUC ↑ | Logloss ↓ | AUC ↑ | Logloss ↓ |
| First-Order | LR [Richardson et al., 2007] | 0.7874 | 0.4551 | 0.7574 | 0.3944 | 0.9449 | 0.2870 | 0.9394 | 0.2407 |
| Second-Order | FM [Rendle, 2010] | 0.8028 | 0.4514 | 0.7618 | 0.3910 | 0.9551 | 0.2328 | 0.9707 | 0.2003 |
| | AFM [Xiao et al., 2017] | 0.8050 | 0.4434 | 0.7598 | 0.3921 | 0.9568 | 0.2224 | 0.9612 | 0.2182 |
| High-Order | HOFM (3rd) [Blondel et al., 2016] | 0.8052 | 0.4456 | 0.7690 | 0.3873 | 0.9573 | 0.2075 | 0.9701 | 0.2006 |
| | NFM [He and Chua, 2017] | 0.8072 | 0.4444 | 0.7708 | 0.3876 | 0.9682 | 0.2064 | 0.9765 | 0.1694 |
| | OPNN [Qu et al., 2016] | 0.8096 | 0.4426 | 0.7718 | 0.3838 | 0.9566 | 0.2333 | 0.9812 | 0.1912 |
| | CIN [Xu et al., 2021a] | 0.8086 | 0.4437 | 0.7739 | 0.3839 | 0.9666 | 0.2066 | 0.9796 | 0.1676 |
| | AFN [Cheng et al., 2020] | 0.8115 | 0.4406 | 0.7748 | 0.3834 | 0.9656 | 0.2440 | 0.9791 | 0.1853 |
| | AutoInt [Song et al., 2019] | 0.8128 | 0.4392 | 0.7725 | 0.3868 | 0.9633 | 0.2155 | 0.9743 | 0.2315 |
| | SAM [Cheng and Xue, 2021] | 0.8135 | 0.4386 | 0.7730 | 0.3863 | 0.9616 | 0.2523 | 0.9801 | 0.1852 |
| Ensemble | DCN [Wang et al., 2017] | 0.8106 | 0.4414 | 0.7749 | 0.3855 | 0.9630 | 0.2162 | 0.9753 | 0.1676 |
| | DeepFM [Guo et al., 2017] | 0.8130 | 0.4389 | 0.7752 | 0.3859 | 0.9625 | 0.2520 | 0.9759 | 0.1993 |
| | xDeepFM [Lian et al., 2018] | 0.8127 | 0.4392 | 0.7747 | 0.3841 | 0.9612 | 0.2186 | 0.9779 | 0.1869 |
| | DRIN [Jun et al., 2022] | 0.8136 | 0.4385 | 0.7729 | 0.3860 | 0.9703 | 0.1998 | 0.9812 | 0.1592 |
| | MaskNet [Wang et al., 2021b] | 0.8137 | 0.4383 | 0.7731 | 0.3859 | 0.9706 | 0.1992 | 0.9819 | 0.1586 |
| | DCN-V2 [Wang et al., 2021a] | 0.8133 | 0.4393 | 0.7737 | 0.3840 | 0.9675 | 0.2131 | 0.9778 | 0.1910 |
| | LightDIL [Zhang et al., 2023] | 0.8138 | 0.4382 | 0.7751 | 0.3841 | 0.9711 | 0.1993 | 0.9834 | 0.1549 |
| | FinalMLP [Mao et al., 2023] | 0.8139 | 0.4380 | 0.7754 | 0.3837 | 0.9713 | 0.1989 | 0.9836 | 0.1546 |
| | FINAL [Zhu et al., 2023] | 0.8141 | 0.4379 | 0.7756 | 0.3831 | 0.9708 | 0.1996 | 0.9830 | 0.1567 |
| | GDCN [Wang et al., 2023] | 0.8144 | 0.4374 | 0.7758 | 0.3829 | 0.9717 | 0.1984 | 0.9844 | 0.1538 |
| Ours | RE-SORT | **0.8163**[**] | **0.4358**[**] | **0.7779**[**] | **0.3806**[**] | **0.9734**[**] | **0.1956**[**] | **0.9865**[**] | **0.1501**[**] |
| | Rel. Imp | 0.0019 | - 0.0016 | 0.0021 | - 0.0023 | 0.0017 | - 0.0028 | 0.0021 | - 0.0037 |

Table 2: Overall performance comparison in the four datasets. The t-test results show that our performance advantage over the previous SOTA method is statistically significant. The improvement is statistically significant with $p < 10^{-2}$ ($\star$: $p < 10^{-2}$, $\star\star$: $p < 10^{-4}$). "Rel. Imp" denotes the relative improvements compared with the previous SOTA method.

datasets. To mitigate overfitting, we implement early stopping based on the AUC attained on the validation dataset. In the D-SR and S-SR, we define the attenuation coefficient $r$ as $\left[\frac{31}{32}, \frac{63}{64}, ...., \frac{2^{M+5}-1}{2^{M+5}}\right]$.

## 4.2 COMPARISON WITH THE SOTA METHODS

We classify twenty competitive approaches into four categories: first-order, second-order, high-order, and ensemble methods. To ensure a fair comparison, we run each method 10 times with random seeds on a single GPU (NVIDIA A100) and report the average testing performance. Then, we conduct two-tailed t-tests [Liu et al., 2019] to compare the performance. As shown in Table 2, on all four datasets, RE-SORT outperforms all the other models in terms of both AUC and LogLoss, which validates its generalization ability. The greatest improvement is observed for the Avazu and Frappe, where RE-SORT exhibits a relative improvement of 0.21% over the second-best-performing GDCN. Previous two-stream methods such as FinalMLP and GDCN exhibit robust performance, while our RE-SORT constructs more effective multilevel features while eliminating spurious correlations and achieves the best performance. It is crucial to emphasize the significance of accurate CTR prediction, particularly in applications with substantial user bases; even a 0.1% AUC increase, although seemingly modest, can have a substantial impact, significantly boosting overall revenue [Cheng et al., 2016, Wang et al., 2017, Ling et al.,

2017]. Moreover, compared to the previous SOTA model (GDCN), RE-SORT is almost 4 times faster, highlighting an additional valuable improvement, which is explained in detail in the following section.

| Model | Avazu | MovieLens | UGC |
|---|---|---|---|
| GDCN | 40 | 1.8 | 105 |
| Baseline [4.1.1] | 35 | 1.4 | 96 |
| FinalMLP | 30 | 1.1 | 87 |
| Baseline + SA [4.1.1] | 25 | 1.0 | 72 |
| LightDIL | 20 | 0.9 | 64 |
| DRIN | 16 | 0.8 | 50 |
| RE-SORT | **10** | **0.4** | **24** |

Table 3: Inference time (second) comparison on the test set.

**Speed comparison.** The rapid inference speed of the RE-SORT is attributed to its MSR structure. The SCE module does not affect the inference speed because it is used only during training. We choose "Baseline"; "Baseline + SA"; and the cutting-edge two-stream GDCN, FinalMLP, Light-DIL, and DRIN methods for comparison. Table 3 demonstrates that our RE-SORT substantially outperforms the previous SOTA methods; it outperforms DRIN, the fastest among these methods, by approximately 90%.

---

The GDCN yields a 0.17% improvement over the previous SOTA approach on Criteo. With its stacked structure, GDCN achieves a 0.04% improvement over the baseline.

| Model | Criteo | | Avazu | | UGC | |
|---|---|---|---|---|---|---|
| | AUC ↑ | Logloss ↓ | AUC ↑ | Logloss ↓ | AUC ↑ | Logloss ↓ |
| Previous SOTA (GDCN) | 0.8144 | 0.4374 | 0.7758 | 0.3829 | 0.8037 | 0.4055 |
| Baseline [4.1.1] | 0.8138 | 0.4387 | 0.7752 | 0.3845 | 0.8033 | 0.4064 |
| Baseline + SA [4.1.1] | 0.8140 | 0.4384 | 0.7753 | 0.3842 | 0.8035 | 0.4059 |
| Baseline + MSR | 0.8152 | 0.4377 | 0.7764 | 0.3827 | 0.8046 | 0.4044 |
| Baseline + SCE | 0.8154 | 0.4363 | 0.7770 | 0.3818 | 0.8053 | 0.4035 |
| Full Approach (MSR + SCE) | **0.8163**[**] | **0.4358**[**] | **0.7779**[**] | **0.3806**[**] | **0.8058**[**] | **0.4027**[**] |

Table 4: Ablation study of the RE-SORT. "Baseline + MSR" denotes that the two-stream MLP network of "Baseline" is replaced by MSR, and the implementation of these models can be found in our code. The last row is our full approach.

## 4.3 ABLATION STUDY

We verify the effectiveness of the MSR and the SCE structures through extensive experiments.

**Improvement achieved with the MSR and SCE.** Table 4 shows that MSR and SCE stably improve the performance of the model, exceeding the "Baseline" by an average of 0.25% in terms of the AUC across all three datasets. To fairly compare the performances of various two-stream structures, the results presented in the table are the best experimental results obtained with different depth combinations for the two streams within their corresponding network structures. Our MSR outperforms the two-stream MLP and two-stream SA structures by an average of 0.12% and 0.11% AUC, respectively. Tables 3 and 4 demonstrate that the proposed MSR structure outperforms MLP and attention-based models in terms of both accuracy and computational efficiency.

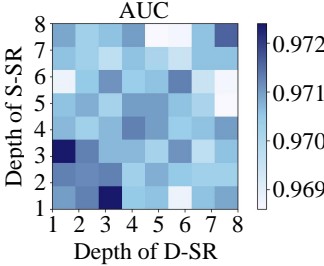

Figure 3: Performance with different MSR depths.

**Two streams in MSR with different depths.** We verify the performance of RE-SORT with different combinations of the S-SR and D-SR streams at various depths. On the five datasets, the best performance is obtained with D-SR and S-SR depths of 3 and 1, respectively. Figure 3 shows a visualization example based on the MovieLens dataset. When the D-SR and S-SR depths are the same (1, 2, and 3), the AUC performance is similar to that of the "Baseline" (0.9712, 0.9713, and 0.9711 vs. 0.9711), which shows that the MSR structure is not able to construct multilevel features with the same stacking depth to achieve improved performance. When the depths of the D-SR and S-SR streams are

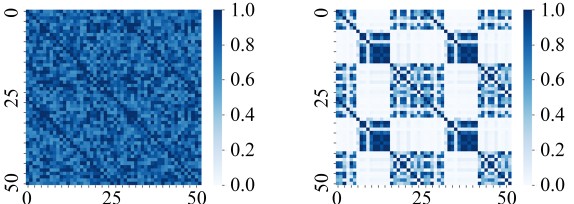

(a) Feature correlation of "Baseline" on Avazu. (b) Feature correlation of SCE on Avazu.

Figure 4: Spurious correlation elimination of RE-SORT.

set to 3 and 1, respectively, the result is 0.12% greater than that of the "Baseline".

**Feature visualization for SCE.** We randomly select variables with 50 dimensions from input features of the CTR predictor learned by "Baseline" and SCE. Figure 4 shows that the correlations among the features are eliminated by SCE, while the model performance is notably improved. To a certain extent, this experiment substantiates that SCE can promote the model to identify critical features and acquire more effective associations with CTR predictions, thus enhancing the generalization ability. Additional experiments, such as a comparative analysis of using different numbers of random Fourier spaces, are reported in the appendix.

## 5 CONCLUSION

We introduce the RE-SORT for CTR prediction. The proposed MSR enables the network to efficiently capture diverse high-order representations. An SCE module is introduced to effectively remove spurious correlations, allowing the model to focus on true causal features. Extensive experiments on four public datasets and our production dataset demonstrate the SOTA performance of RE-SORT, which advances the current methods and offers valuable insights for future research and applications.

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

# APPENDIX

# A   ADDITIONAL DEFINITIONS, ASSUMPTIONS, AND THEOREMS FOR THE SCE MODULE

## A.1   SAMPLE REWEIGHTING

Assumption 1 indicates that the correlation between $\mathrm{FM_{rob}}$ and $g(\mathrm{FM_{rob}})$ has lower variation than that between $\mathrm{FM_{fal}}$ and $g(\mathrm{FM_{rob}})$ when subjected to parameter variance.

**Assumption 1.** *For all $P'$ uniformly sampled from $\mathcal{P}$, the following condition holds: $Var[\mathbb{E}_{P'}(\mathrm{FM_{rob}}g(\mathrm{FM_{rob}}))] < Var[\mathbb{E}_{P'}(\mathrm{FM_{fal}}g(\mathrm{FM_{rob}}))]$.*

Theorem 1 He et al. [2023] demonstrates that when we apply sample weighting even in a random way, the estimation of $\beta_{\mathrm{FM_{rob}}}$ is more robust and exhibits less variation in comparison to $\beta_{\mathrm{FM_{fal}}}$.

**Theorem 1.** *Let $\tilde{\beta}_{\mathrm{FM_{rob}}}$ and $\tilde{\beta}_{\mathrm{FM_{fal}}}$ be the components of $\tilde{\beta}$ corresponding to $\mathrm{FM_{rob}}$ and $\mathrm{FM_{fal}}$ respectively. Under Assumption 3.5.1, we have $Var(\tilde{\beta}_{\mathrm{FM_{rob}}}) < Var(\tilde{\beta}_{\mathrm{FM_{fal}}})$.*

We define the minimal robust features set.

**Definition 1.** *minimal robust features set: A minimal robust features set of predicting $Y$ under training distribution $P^{train}$ is any subset $\mathrm{FM_{rob}}$ of FM satisfying the following equation, and no proper subsets of $\mathrm{FM_{rob}}$ satisfies it:*

$$\mathbb{E}_{P^{train}}[Y|\mathrm{FM_{rob}}] = \mathbb{E}_{P^{train}}[Y|\mathrm{FM}]. \tag{19}$$

*It can also be formulated as:*

$$Y \perp \mathrm{FM_{fal}}|\mathrm{FM_{rob}},$$

*where $Y \perp \mathrm{FM_{fal}}$ means $Y$ and $\mathrm{FM_{fal}}$ are statistically independent of each other.*

It has been proven that $\mathrm{FM_{rob}}$ is the minimal and optimal predictor if and only if $\mathrm{FM_{rob}}$ is the minimal robust feature [Xu et al., 2022]. Therefore, we intend to capture the minimal robust feature set $\mathrm{FM_{rob}}$ for robust prediction. We denote the robust feature set as the minimal robust feature set for simplicity.

**Definition 2.** *Sample weighting function: Let $\mathcal{W}$ be the set of sample weighting functions that satisfy:*

$$\mathcal{W} = \{w : \mathrm{FM} \to R^+ | E_{P^{train}(\mathrm{FM})}[w(\mathrm{FM})] = 1\}. \tag{20}$$

Then $\forall w \in \mathcal{W}$, the corresponding weighted distribution is $\widetilde{P}_w(\mathrm{FM}, Y) = w(\mathrm{FM})P^{train}(\mathrm{FM}, Y)$, $\widetilde{P}_w$ is well defined with the same support of $P^{train}$. What we want is actually $W_\perp$: the subset of W in which FM are mutually independent.

**Lemma 1.** *Existence of a "debiased" weighting function [Zhou et al., 2022]. Consider $FM=[\mathrm{FM_{rob}}, \mathrm{FM_{fal}}]$. We want to fit a linear model $\theta^T\mathrm{FM}$ to predict $y$. Given infinite data in the training dataset $D_{train}$, there exists a weight function $w \in \mathcal{W}$, i.e.,*

$$w(\mathrm{FM}, y) = \frac{\mathbb{P}(\mathrm{FM_{rob}}, y)\mathbb{P}(\mathrm{FM_{fal}})}{\mathbb{P}(\mathrm{FM_{rob}}, \mathrm{FM_{fal}}, y)}, \tag{21}$$

*such that the solution satisfies that*

$$\theta^*(w) = \overline{\theta} = [\overline{\theta}_{\mathrm{FM_{rob}}}; 0], \tag{22}$$

*where $\overline{\theta}_{\mathrm{FM_{rob}}}$ is the optimal model that merely uses $\mathrm{FM_{rob}}$, i.e.,*

$$\overline{\theta}_{\mathrm{FM_{rob}}} := \underset{\theta \in R^{d_{\mathrm{FM_{rob}}}}}{arg\,min} \; \mathbb{E}[(y - \theta^T\mathrm{FM_{rob}})^2]. \tag{23}$$

**Theorem 2.** *No matter whether data generation is linear or nonlinear, robust features can be almost perfectly selected if conducting weighted least squares (WLS) using the weighting function in $\mathcal{W}_\perp$ [Xu et al., 2021b].*

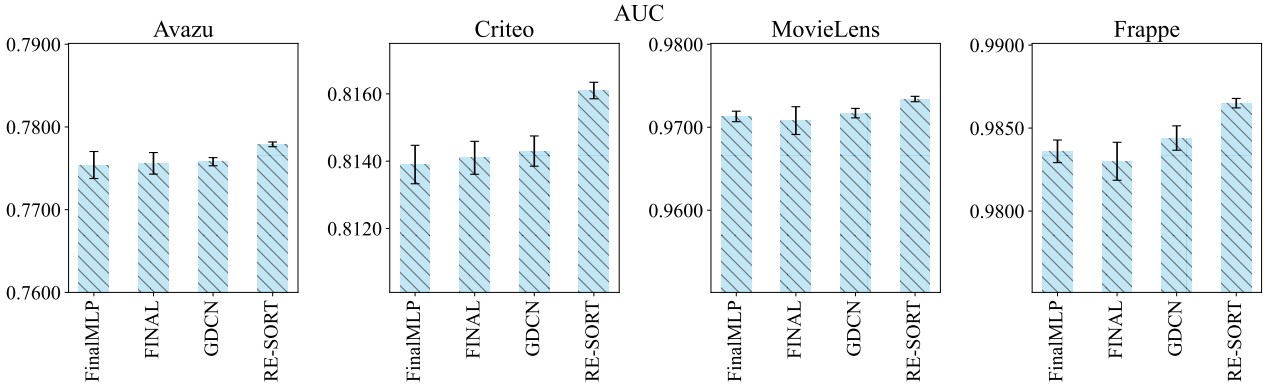

Figure 5: Variance comparison of different SOTA models.

## A.2 SCE

Here, we use $X$ and $Y$ to denote random variables, the corresponding RKHS is denoted by $\mathcal{H}_X$ and $\mathcal{H}_Y$. We define the cross-covariance operator $\sum_{XY}$ from $\mathcal{H}_Y$ to $\mathcal{H}_X$:

$$< h_X, \sum_{XY} h_Y >$$
$$= \mathbb{E}_{XY}[h_X(X)h_Y(Y)] - \mathbb{E}_X[h_X(X)]\mathbb{E}_Y[h_Y(Y)] \tag{24}$$

for all $h_X \in \mathcal{H}_X$ and $h_Y \in \mathcal{H}_Y$. Then, the independence can be determined by the following proposition [Fukumizu et al., 2007]:

**Proposition 1.** *If the product $k_X k_Y$ is characteristic, $\mathbb{E}[k_X(X,X)] < \infty$ and $\mathbb{E}[k_Y(Y,Y)] < \infty$, we have that $\mathbb{E}_{XY} = 0$ if and only if $X \perp Y$.*

The Hilbert-Schmidt independence criterion (HSIC) [Gretton et al., 2008], which necessitates that the squared Hilbert-Schmidt norm of $\sum_{XY}$ be equal to zero, can be utilized as a criterion for supervising the elimination of spurious correlations [Bahng et al., 2020].

# B ADDITIONAL EXPERIMENT RESULTS

**Analysis of the performance variance.** We analyze the performance variance of our model in comparison to that of three representative SOTA models: FinalMLP [Mao et al., 2023], FINAL [Zhu et al., 2023], and GDCN [Wang et al., 2023]. We carried out 10 experiments for each model on each dataset. As depicted in Figure 5, our model consistently outperforms and exhibits the greatest stability across different datasets compared to the other methods.

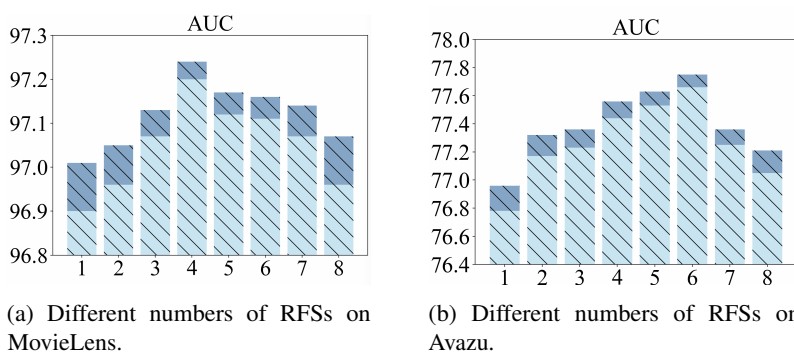

(a) Different numbers of RFSs on MovieLens.

(b) Different numbers of RFSs on Avazu.

Figure 6: Comparison of using fixed number (light blue) and decreasing numbers (dark blue) of RFS.

**Different numbers of random Fourier spaces (RFSs).** We analyze the influence of using different numbers of RFSs. In Figure 6, the light blue columns show the different AUCs obtained using different numbers of fixed RFSs (1∼8) on MovieLens and Avazu. Through extensive experiments, we further find that compared to utilizing a fixed number of RFSs, using decreasing numbers of RFSs in each epoch further yields improved performance. The dark blue columns show the performance achieved with decreasing numbers of RFSs, and the horizontal axis represents the maximum value (the number of RFSs uniformly decreases from the maximum value to 1 in each epoch).

**Feature correlation visualizations of the S-SR and D-SR streams.** We randomly chose 50 feature variables from both the Avazu and Frappe datasets. The feature correlation maps of the S-SR and D-SR streams are depicted in Figure 7. As illustrated in the figure, noticeable differences exist between the two feature correlation maps across both datasets.

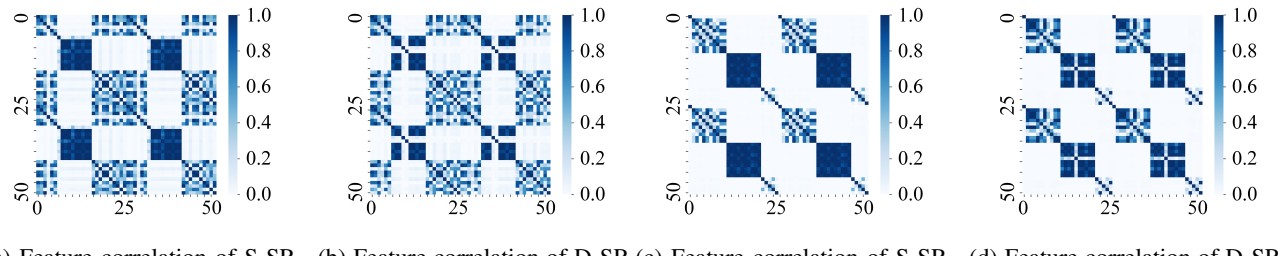

(a) Feature correlation of S-SR on the Avazu dataset.

(b) Feature correlation of D-SR on the Avazu dataset.

(c) Feature correlation of S-SR on the Frappe dataset.

(d) Feature correlation of D-SR on the Frappe dataset.

Figure 7: Feature visualization of D-SR and S-SR.

## C   RELATED METHODS

We classify the related CTR prediction methods into four types:

1. **First-Order**: LR [Richardson et al., 2007]. It models both first-order and second-order feature interactions.

2. **Second-Order**: FM [Rendle, 2010], AFM [Xiao et al., 2017]. They model both first-order and second-order interactions.

3. **High-Order**: HOFM [Blondel et al., 2016], NFM [He and Chua, 2017], OPNN [Qu et al., 2016], CIN [Xu et al., 2021a], AutoInt [Song et al., 2019], AFN [Cheng et al., 2020], SAM [Cheng and Xue, 2021]. They can model interactions higher than second-order.

4. **Ensemble**: DCN [Wang et al., 2017], DeepFM [Guo et al., 2017], xDeepFM [Lian et al., 2018], DRIN [Jun et al., 2022], MaskNet [Wang et al., 2021b], DCN-V2 [Wang et al., 2021a], LightDIL [Zhang et al., 2023], FINAL [Zhu et al., 2023], FinalMLP [Mao et al., 2023], DELTA [Zhu et al., 2024], GDCN [Wang et al., 2023]. These models adopt parallel or stacked structures to integrate different feature interaction methods.