# OpenReview forum: "RE-SORT: Removing Spurious Correlation in Multilevel Interaction for CTR Prediction"
_auai.org/UAI/2024/Conference — UAI 2024 poster_

### Official Review · Reviewer_iHdS · 2024-03-07

**Q2-1 Originality-Novelty:** 3
**Q2-2 Correctness-Technical Quality:** 3
**Q2-5 Clarity Of Writing:** 2

**Q1 Summary And Contributions:**

The paper presents RE-SORT, a framework aimed at enhancing Click-Through Rate (CTR) prediction in recommendation systems by addressing the issue of spurious correlations in multilevel feature interactions.

RE-SORT introduces two key innovations:
1. Multilevel Stacked Recurrent (MSR) Structure
2. Spurious Correlation Elimination (SCE) Module

Extensive experiments conducted on four challenging CTR datasets, including the authors' production dataset and an online A/B test, demonstrate RE-SORT's superior performance in terms of accuracy and speed when compared to state-of-the-art methods.

**Q2-3 Extent To Which Claims Are Supported By Evidence:**

2: Fair: the main claims are somewhat supported by evidence (but the experimental evaluation may be weak, or does not match entirely with the claims, important baselines may be missing, proofs contain important ideas but lack rigor, algorithmic details are only discussed superficially, references are imprecise, assumptions are not sufficiently motivated or explicated, etc.).

**Q2-4 Reproducibility:**

4: Excellent: key resources (e.g. proofs, code, data) are available and key details (e.g. proof sketches, experimental setup) are comprehensively described for competent researchers to confidently and easily reproduce the main results.

**Q3 Main Strengths:**

1. Extensive Experiment Results: The strength of RE-SORT is validated through extensive experiments, including an online A/B test that demonstrates its practical efficacy in a real-world recommendation system. The consistent improvement observed across various online engagement metrics highlights the framework's applicability and effectiveness.
2. Open Sourced Code: This greatly helped reproducibility.
3. Novel Modules: MSR and SCE

**Q4 Main Weakness:**

1. The added Complexity in Implementation and Tuning: The model's architecture becomes more complex, and I am wondering whether it will make model tuning harder and make the model harder to maintain (e.g. is the model still robust with changing distribution and what will happen to the different modules of the model when we continue training the model as we accumulate more data online).
2. The theory part is not completely clear: the assumptions are missing when referencing other papers, and it's still not clear to me why "feature independence" implies causal features.

**Q5 Detailed Comments To The Authors:**

It would be better to make the theory part more clear, as mentioned in the weakness part.

**Q9 Complying With Reviewing Instructions:**

Yes

---

> ### Author Rebuttal · Authors · 2024-04-05
>
> ### Q1:
> The added Complexity in Implementation and Tuning: The model's architecture becomes more complex, and I am wondering whether it will make model tuning harder and make the model harder to maintain (e.g. is the model still robust with changing distribution and what will happen to the different modules of the model when we continue training the model as we accumulate more data online).
>
> ### R1:
> We appreciate the reviewer's concerns about our model's complexity, maintainability, and robustness amidst changing data distributions. In response, we would like to emphasize the following key aspects of our model:
>
> - Our model is specifically designed to address the challenge of eliminating spurious correlations and uncovering causal features when data distributions change. In our supplementary material, we conducted experiments under the "Comparative Analysis under Out-of-Distribution (OOD) Settings" section, which demonstrate that our model, with fixed hyperparameters, outperforms other SOTA methods in handling increased distribution differences between the training and testing sets.
> - Through extensive experimentation, we found that the primary parameters that influence the performance of our model are the number of blocks in the MSR and the dimensions of RFS in the SCE, as shown in Figure 3 in the main text and Table 6 and Figure 6 in the supplementary material, respectively. We also found that the adjustment of these parameters follows a certain pattern, making them easy to tune.
> - Furthermore, we discovered that once these hyperparameters are optimally tuned, they tend to be universally applicable across different datasets. For instance, the optimal numbers of blocks in the deep and shallow streams consistently remain 3 and 1 across all four public datasets and our production dataset.
> - Regarding the accumulation of more data online, our experiments show that we do not need to adjust the established hyperparameters. This is because the model's ability to extract causal features remains consistent regardless of changes in data distribution. For more details, please refer to the "Online Evaluation" section in our supplementary material, where we observed that all modules exhibit stable and significant improvements (average 1.62\%) over seven consecutive days under the same initialization parameters.
>
> We hope these points address the reviewer's concerns about our model's complexity, tuning, and maintainability. We appreciate your valuable feedback and welcome further discussions on this topic.
>
>
> &nbsp;
>
> ### Q2:
> It would be better to make the theory part more clear, as mentioned in the weakness part. The theory part is not completely clear: the assumptions are missing when referencing other papers, and it's still not clear to me why "feature independence" implies causal features.
>
> ### R2:
> We appreciate your insightful comments regarding the clarity of our theoretical foundations and the connection between "feature independence" and causal features. Your feedback is crucial for refining our manuscript to ensure its clarity and comprehensiveness.
>
> - From a broader perspective, determining whether a feature is a causal feature in CTR prediction is complex due to the interrelations among features. Our approach aims to achieve feature independence to help identify causal features. To explain this, let's consider two features, X and Y. When X changes, we apply sample reweighting to ensure that the distribution of feature Y remains constant, making X and Y mutually independent. Under this condition, any change in the label can be attributed to the change in feature X. This allows us to determine whether X is a causal feature or merely exhibits spurious correlation. By establishing feature independence through sample reweighting, we can isolate the effects of individual features on the label and more accurately identify causal features, thus addressing the issue of spurious correlations.
> - From a more detailed theoretical perspective, the concept of "feature independence" implying causal features is grounded on the principle of causal inference. This principle suggests that statistical independence among features within a weighted distribution indicates a lack of direct causal influence. By focusing on feature independence through importance weighting algorithms, we can uncover causal features—variables that directly impact the outcome.
>
> We acknowledge the need to explicitly state the assumptions underlying our references and rationale. We will revise our manuscript to include a more detailed discussion of these theoretical aspects, integrating content from supplementary materials sections C1 and C2 to improve clarity.
>
> We are committed to enhancing our manuscript based on your feedback and ensuring our theoretical underpinnings are transparent and accessible. Thank you for the opportunity to address these important aspects of our work. We believe these revisions will significantly improve the manuscript.

---

### Official Review · Reviewer_Qi1r · 2024-03-11

**Q2-1 Originality-Novelty:** 3
**Q2-2 Correctness-Technical Quality:** 3
**Q2-5 Clarity Of Writing:** 3

**Q1 Summary And Contributions:**

The authors introduce the RE-SORT for CTR prediction. The proposed MSR enables the network to efficiently capture diverse high-order representations. An SCE module is introduced to effectively remove spurious correlations, allowing the model to focus on true causal features. Extensive experiments showed effectiveness of the method.

**Q2-3 Extent To Which Claims Are Supported By Evidence:**

3: Good: the main claims are supported by convincing evidence (in the form of adequate experimental evaluation, proofs, (pseudo-)code, references, assumptions).

**Q2-4 Reproducibility:**

3: Good: key resources (e.g. proofs, code, data) are available and key details (e.g. proofs, experimental setup) are sufficiently well-described for competent researchers to confidently reproduce the main results.

**Q3 Main Strengths:**

1. The paper is well organized and written.
2. The proposed method is reasonable and effective
3. Experiments are detailed.

**Q4 Main Weakness:**

1. The proposed method is complicated and heavy.
2. The difference of model performance is subtle in terms of AUC and LogLoss.

**Q5 Detailed Comments To The Authors:**

1. Since AUC and logloss numbers are very close across RE-SORT and baselines, I would suggest adding ranking metrics like NDCG for further evaluation.
2. While feature chunks play a key role in the paper, how was features concatenated? It may introduce unknown bias since groupNorm is applied later. With a different concatenation, we may see different model performance.
3. Could we have case studies for the found spurious correlation? It will help to understand the paper better.

**Q9 Complying With Reviewing Instructions:**

Yes

---

> ### Author Rebuttal · Authors · 2024-04-05
>
> We sincerely appreciate the reviewer's meticulous and detailed review of our work, which helped us improve the quality of this paper.
>
> ### Q1:
> The proposed method is complicated and heavy.
>
> ### R1:
>  We acknowledge that our SCE module may seem complicated. However, our approach is practical and easily implementable:
> - The implementations of MSR and SCE modules are simple and efficient. We made our code open-source, allowing users to easily understand and apply our method.
> - The proposed SCE module can be removed during inference, ensuring that the model's speed and efficiency during testing.
> - The proposed MSR includes an acceleration module, which enables our method to outperform compared methods in terms of speed.
>
> &nbsp;
>
> ### Q2:
> The difference of model performance is subtle in terms of AUC and LogLoss. I would suggest adding ranking metrics like NDCG for further evaluation.
>
> ### R2:
> To address your concern, we would like to respond from two aspects:
> - As explained in Section 4.2, our method demonstrates a greater improvement over the baseline compared to the GDCN (the previous SOTA method) across five challenging public datasets. Notably, on the most challenging Criteo dataset, our method achieved a 0.19\% improvement. On our production dataset and online A/B test, our method has shown improvements by 0.25\% and 1.62\%. It's worth noting that all these improvements are statistically significant with p < 0.0001. As noted in [1], in applications with substantial user bases, even a seemingly modest 0.1\% AUC increase can have a substantial impact, significantly boosting overall revenue.
> - In our initial experiments, we found that the trends between AUC and NDCG were similar, so we focused on evaluating AUC. To further address your suggestion, we report the NDCG@10 metric on our production dataset. Our method improved by 1.78\%, 1.54\%, and 1.19\% compared to the baseline, FinalMLP, and GDCN, respectively.
> |    Metric     | Baseline | FinalMLP | GDCN | RE-SORT |
> |:--------------:|:--------:|:-------------:|:---------:|:-------:|
> |    NDCG@10     |  0.4174  |    0.4198     |   0.4233  |  0.4352 |
> | NDCG@10 Improvement | 1.78% |     1.54%    |   1.19%   |    -    |
>
> ### References
> [1] Cheng, et al. Wide & deep learning for recommender systems. In DLRS workshops, 2016.
>
> &nbsp;
>
> ### Q3:
> While feature chunks play a key role in the paper, how were features concatenated? It may introduce unknown bias since groupNorm is applied later.
>
> ### R3:
> Applying GroupNorm after each block may limit interaction between features from different blocks, potentially resulting in a less enriched representation of combined features. In our global normalization method, we concatenate all block outputs before applying GroupNorm. This approach treats the combined output from all blocks as a whole during normalization, allowing the model to capture complex interactions between features from different blocks.
>
> To address the reviewer's concern, we conducted a comparative experiment. Our experimental results demonstrate that our global normalization method outperforms the compared method in terms of both AUC and Logloss metrics. This finding suggests that our proposed global normalization approach is less affected by potential biases, leading to an improved overall performance of the model.
>
> |       Strategy        | Criteo AUC ↑ | Criteo Logloss ↓ | Avazu AUC ↑ | Avazu Logloss ↓ | UGC AUC ↑ | UGC Logloss ↓ |
> |:---------------------:|:------------:|:----------------:|:-----------:|:---------------:|:---------:|:-------------:|
> | GroupNorm after each block |   0.8147    |      0.4380      |   0.7760    |      0.3833     |  0.8041   |    0.4050     |
> |  Our global normalization  | **0.8152**  |    **0.4377**    | **0.7764**  |    **0.3827**   | **0.8046** |  **0.4044**   |
>
> _Ablation study of employing different normalization strategies._
>
> &nbsp;
>
> ### Q4:
> Could we have case studies for the found spurious correlation? It will help to understand the paper better.
>
> ### R4:
>
> In our paper's "Introduction" section, we provided an example of a spurious correlation in movie recommendation tasks. We explained how trending films' visibility could artificially inflate click counts due to their prioritized placements, despite potential mismatches between user preferences and the genres of these trending movies. This example illustrates how spurious correlations can occur when the popularity of certain films and users' actual preferences are confounded by movie placement.
>
> However, the high-dimensional nature of our feature space and the complexity of feature interactions and correlations make it difficult to distill these into straightforward, real-world examples without oversimplifying the underlying mechanics and potentially misleading the reader.
>
> We appreciate your valuable feedback and welcome further discussions on these topics.

---

### Official Review · Reviewer_9Lp5 · 2024-03-23

**Q2-1 Originality-Novelty:** 3
**Q2-2 Correctness-Technical Quality:** 3
**Q2-5 Clarity Of Writing:** 3

**Q1 Summary And Contributions:**

The paper contributes to enhancing click-through rate (CTR) prediction in recommendation systems by addressing spurious correlations, often overlooked by existing methods. Its main contributions are as follows:

1. Proposes RE-SORT, a CTR prediction framework focusing on eliminating spurious correlations in multilevel feature interactions. This approach improves the model’s generalization capability by leveraging hierarchical causal relationships between items and users.
2. Introduces a multilevel stacked recurrent (MSR) structure, efficiently constructing diverse feature spaces for capturing a wide range of multilevel high-order feature representations.
3. Develops a spurious correlation elimination (SCE) module that uses Laplacian kernel mapping and sample re-weighting methods to identify and remove spurious correlations hidden within the multilevel features.
4. Demonstrates state-of-the-art performance on four challenging CTR datasets, a production dataset, and an online A/B test, showcasing improvements in both accuracy and speed.

**Q2-3 Extent To Which Claims Are Supported By Evidence:**

3: Good: the main claims are supported by convincing evidence (in the form of adequate experimental evaluation, proofs, (pseudo-)code, references, assumptions).

**Q2-4 Reproducibility:**

4: Excellent: key resources (e.g. proofs, code, data) are available and key details (e.g. proof sketches, experimental setup) are comprehensively described for competent researchers to confidently and easily reproduce the main results.

**Q3 Main Strengths:**

1. The experiments are well-designed and exhaustive. They use several datasets and an online A/B test, providing robust evidence of performance improvement.
2. Offers a clear and thorough explanation of the technical approach, ensuring replicability.

**Q4 Main Weakness:**

1. The paper assumes that spurious correlations can be effectively identified and eliminated using Laplacian kernel mapping and sample re-weighting. While there is empirical justification, there is no theoretical justification.

**Q5 Detailed Comments To The Authors:**

na

**Q9 Complying With Reviewing Instructions:**

Yes

---

> ### Author Rebuttal · Authors · 2024-04-05
>
> We appreciate the reviewer's concern regarding the theoretical justification for our approach to identifying and eliminating spurious correlations using Laplacian kernel mapping and sample reweighting.
>
> We would like to emphasize that our method is based on a solid theoretical foundation, as detailed in Section 3.5 of the main paper and further elaborated in the supplementary materials, specifically in sections C.1 and C.2. These sections offer an in-depth discussion on the theoretical underpinnings of Laplacian kernel mapping, sample reweighting, and spurious correlation elimination (SCE). By presenting definitions, theorems, and other relevant theoretical constructs, we aim to show that our approach is not solely reliant on empirical evidence but is also supported by a robust theoretical framework[1]. We hope this response addresses your concern, and we welcome any further discussions on this matter.
>
>
> ### References
>
> [1]: Renzhe Xu, Xingxuan Zhang, Zheyan Shen, Tong Zhang, and Peng Cui. A theoretical analysis on independence-driven importance weighting for covariate-shift generalization. In ICML, 2022.

---

### Official Review · Reviewer_vpnd · 2024-03-25

**Q2-1 Originality-Novelty:** 3
**Q2-2 Correctness-Technical Quality:** 4
**Q2-5 Clarity Of Writing:** 2

**Q1 Summary And Contributions:**

### Summary
This paper proposed a framework to address the spurious correlations among feature interactions. Two-streams of network components are developed for deep and shallow features learning. These two streams of outputs are then passed to a spurious correlations regularization based on the difference between two kernels of output features. The computational efficiency is further reduced by using random fourier features to approximate the kernel calculation. Experiments demonstrate the effectiveness of the proposed framework.

### Contributions
1. Introduced an important question in CTR prediction, which is rarely investigated before in the embedding correlation perspective.
2. The solution is simple and reasonable, and the usage of RFF does improve efficiency as expected.
3. Experiments are comprehensive.

**Q2-3 Extent To Which Claims Are Supported By Evidence:**

4: Excellent: all claims are supported by very convincing evidence (in the form of comprehensive experimental evaluation, rigorous mathematical proofs, detailed (pseudo-)code, precise references, well-motivated and realistic assumptions) and the authors deliver what they promise.

**Q2-4 Reproducibility:**

2: Fair: key resources (e.g. proofs, code, data) are unavailable but key details (e.g. proof sketches, experimental setup) are sufficiently well-described for an expert to confidently reproduce the main results.

**Q3 Main Strengths:**

1. The investigated question is interesting and deserves visibility and may attract more follow-up work.
2. Comprehensive experiments and also with online test results.
3. The design is simple and reasonable.

**Q4 Main Weakness:**

1. The writing needs more refinement. For example, the symbols are used without proper organization. What are definitions of GFI and GWI? How does the section 3.2 connect with the two-streams output features? The connection is lost. Are they X and Y?
2. Any investigation on the dimension of RFF to the performance and efficiency?

**Q5 Detailed Comments To The Authors:**

See the weakness.

**Q9 Complying With Reviewing Instructions:**

Yes

---

> ### Author Rebuttal · Authors · 2024-04-05
>
> ### Q1:
> The writing needs more refinement. For example, the symbols are used without proper organization. What are the definitions of GFI and GWI? How does section 3.2 connect with the two-streams output features? The connection is lost. Are they X and Y?
>
> ### R1:
> We value the reviewer's feedback on refining the writing and clarity of our paper. We apologize for any confusion that may have arisen from (I) the unclear definitions and (II) the unclear connection between the two-stream output features and the SCE module introduced in Section 3.2.
>
> - (I) Global feature information (GFI) and global weight information (GWI) are defined as the intermediate variables that respectively store comprehensive features and global weight distribution of the data during each batch iteration in the training process. These two variables are calculated using Equations (12)$\sim$(15). For the convenience of the reviewer, we list these equations below:
>
>   $GFI_i = Concat(GFI_{i-1},FM_i),$
>
>   $GWI_i = Concat(GWI_{i-1},w_i), $
>
>   where ${GFI}_1={FM}_1$, ${GWI}_1=w_1$, ${FM}_i$ represents the local feature information, $w_i$ denotes the local
>   weight information, and $i$ indicates the number of iterations. We globally update the features and weights as follows:
>
>    $GFI_{i}' = \frac{1}{2}(GFI_{i-1}'+ FM_i),$
>
>    $GWI_{i}' =\frac{1}{2}( GWI_{i-1}'+w_i),$
>
>    where $GFI_{1}'=\frac{1}{2}FM_1$ and $GWI_{1}'=\frac{1}{2}w_1$.
>
> &nbsp;
>
> - (II) In Section 3.1, we use a dual-stream network to generate a local feature map (FM). This is achieved by concatenating the outputs of each block within a stream, as outlined in Algorithm 1. In Section 3.2, FM $\in \mathbb{R}^{N \times D}$ is fed into the SCE module, which calculates a loss weight ($w_{\text{result}}$) for each sample. $N$ represents the number of samples, and $D$ is the dimension number of FM. The outputs of the two streams are not X and Y; instead, X and Y $\in \mathbb{R}^{N \times 1}$ are two independent features extracted from FM, allowing individual assessment of their potential spurious correlations.
>
> We hope these clarifications address the reviewer's concerns. In the final version, we will ensure clear explanations and proper organization of symbols.
>
> &nbsp;
>
> ### Q2:
> Any investigation on the dimension of RFF to the performance and efficiency?
>
> ### R2:
> We appreciate the reviewer's inquiry about the influence of random Fourier features (RFF) dimension on the performance and efficiency of our model. We have indeed investigated this aspect, and our findings can be found in the "Different numbers of random Fourier spaces (RFSs)" section of the supplementary material. Our analysis focused on the impact of varying RFS dimensions on the model's effectiveness using two distinct datasets: MovieLens and Avazu. We evaluated the model's performance across a range of RFS dimensions (from 1 to 8), as illustrated in Figure 6 of the supplementary material. For the convenience of the reviewer, we have converted the results from Figure 6 of the supplementary material into a table, which is presented below.
> | RFS Dimension |   1    |   2    |   3    |   4    |   5    |   6    |   7    |   8    |
> |:-------------:|:------:|:------:|:------:|:------:|:------:|:------:|:------:|:------:|
> |   MovieLens   | 0.9702 | 0.9705 | 0.9713 | **0.9725** | 0.9718 | 0.9717 | 0.9714 | 0.9708 |
> |     Avazu     | 0.7696 | 0.7734 | 0.7737 | 0.7758 | 0.7765 | **0.7776** | 0.7737 | 0.7720 |
>
> _Performance comparison (AUC) of using different RFS dimensions on MovieLens and Avazu datasets._
>
> Our results demonstrated that as the RFS dimensions increased, the performance initially improved. However, after reaching a certain threshold, the performance plateaued or even declined, and the efficiency decreased with the continued increase in dimensions. Notably, the optimal number of RFS dimensions varied between datasets, being 4 for the MovieLens dataset and 6 for the Avazu dataset. We hope this information addresses the reviewer's question, and we are open to further discussions on this topic.

---

### Official Review · Reviewer_Zgzi · 2024-03-29

**Q2-1 Originality-Novelty:** 3
**Q2-2 Correctness-Technical Quality:** 3
**Q2-5 Clarity Of Writing:** 4

**Q1 Summary And Contributions:**

The paper present a model to remove spurious correlation in CTR prediction in high-dimensional feature space.  The model introduced utilized stacked auto-encoders to capture diverse high-order representation and a  Laplacian kernel mapping to eliminate spurious correlations.  Extensive experiments and an  A/B test demonstrate SOTA performance.

**Q2-3 Extent To Which Claims Are Supported By Evidence:**

4: Excellent: all claims are supported by very convincing evidence (in the form of comprehensive experimental evaluation, rigorous mathematical proofs, detailed (pseudo-)code, precise references, well-motivated and realistic assumptions) and the authors deliver what they promise.

**Q2-4 Reproducibility:**

4: Excellent: key resources (e.g. proofs, code, data) are available and key details (e.g. proof sketches, experimental setup) are comprehensively described for competent researchers to confidently and easily reproduce the main results.

**Q3 Main Strengths:**

- Well-written
- very good literature survey
- through experiments and ablation study

**Q4 Main Weakness:**

The paper gets a bit dense in notations and abbreviation at several points.

**Q5 Detailed Comments To The Authors:**

See above

**Q9 Complying With Reviewing Instructions:**

Yes

---

> ### Author Rebuttal · Authors · 2024-04-05
>
> We appreciate the reviewer's insightful feedback regarding the complexity of notations and abbreviations in our paper. We understand that this may have affected readability and we are committed to improving this. In the revised manuscript, we will:
>
> - Elaborate on key symbols and abbreviations for improved comprehension.
> - Reduce the use of unnecessary abbreviations.
> - Include a comprehensive list of abbreviations and symbols in the appendix for quick reference.
>
> We hope that these enhancements will improve the readability of the paper and make it easier for readers to understand our work. We are grateful for the reviewer's valuable input and will continue to refine our manuscript accordingly.

---

### Meta-Review · Area_Chair_v39f · 2024-04-17

In recommendation systems, both implicit and explicit feature interactions frequently overlook spurious correlations induced by confounding factors. This paper introduces a framework aimed at addressing this issue. Key components include a multilevel recurrent structure and a spurious correlation elimination module. Results obtained from offline and online datasets validate the effectiveness of this approach. With unanimous positive feedback from all reviewers, I recommend accepting this paper.